# The Challenge of Overcoming Antibiotic Resistance in Carbapenem-Resistant Gram-Negative Bacteria: “Attack on Titan”

**DOI:** 10.3390/microorganisms11081912

**Published:** 2023-07-27

**Authors:** Giuseppe Mancuso, Silvia De Gaetano, Angelina Midiri, Sebastiana Zummo, Carmelo Biondo

**Affiliations:** Department of Human Pathology, University of Messina, 98125 Messina, Italy; sdegaetano6@gmail.com (S.D.G.); amidiri@unime.it (A.M.); zummo@unime.it (S.Z.); cbiondo@unime.it (C.B.)

**Keywords:** CRE, CRAB, CRPA, multidrug resistance, alternative strategies

## Abstract

The global burden of bacterial resistance remains one of the most serious public health concerns. Infections caused by multidrug-resistant (MDR) bacteria in critically ill patients require immediate empirical treatment, which may not only be ineffective due to the resistance of MDR bacteria to multiple classes of antibiotics, but may also contribute to the selection and spread of antimicrobial resistance. Both the WHO and the ECDC consider carbapenem-resistant Enterobacteriaceae (CRE), carbapenem-resistant *Pseudomonas aeruginosa* (CRPA), and carbapenem-resistant *Acinetobacter baumannii* (CRAB) to be the highest priority. The ability to form biofilm and the acquisition of multiple drug resistance genes, in particular to carbapenems, have made these pathogens particularly difficult to treat. They are a growing cause of healthcare-associated infections and a significant threat to public health, associated with a high mortality rate. Moreover, co-colonization with these pathogens in critically ill patients was found to be a significant predictor for in-hospital mortality. Importantly, they have the potential to spread resistance using mobile genetic elements. Given the current situation, it is clear that finding new ways to combat antimicrobial resistance can no longer be delayed. The aim of this review was to evaluate the literature on how these pathogens contribute to the global burden of AMR. The review also highlights the importance of the rational use of antibiotics and the need to implement antimicrobial stewardship principles to prevent the transmission of drug-resistant organisms in healthcare settings. Finally, the review discusses the advantages and limitations of alternative therapies for the treatment of infections caused by these “titans” of antibiotic resistance.

## 1. Introduction

Growing antimicrobial resistance (AMR) poses a major threat to human and animal health [1]. In 2019, the US Centers for Disease Control and Prevention (CDC) published a report showing that antimicrobial resistance causes more than 2.8 million infections and 35,000 deaths in the US every year [2]. The steady rise of AMR is truly alarming, and it is predicted that if this trend is not halted, 10 million people could die from multidrug-resistant diseases by 2050, at an annual cost of well over USD 1 trillion [3]. This makes antimicrobial resistance a leading cause of death, behind only ischemic heart disease and stroke [4]. Over the past 25 years, AMR has been recognized as a serious threat to global health, as evidenced by several high-level policy initiatives, such as the European Antimicrobial Surveillance System established in 1998, and more recently the adoption of the Global Action Plan (GAP) by the WHO in 2015 as a blueprint for combating AMR [5,6]. To combat AMR, the action plan provides member states and other international stakeholders with 83 recommendations grouped into five main objectives. It emphasizes the importance of all member states developing effective national antimicrobial resistance strategies by 2017 [7]. Although 140 countries have developed their own national action plans, implementation is at different stages in different countries [7]. Moreover, only 77 of the 140 plans have been officially approved and are available in the WHO library [7]. In 2016, the USA reported the first case of a patient with an infection that was resistant to all available antimicrobials [8]. Carbapenem-resistant *Klebsiella pneumoniae* carrying the New Delhi metallo-β-lactamase gene (NDM-1) was isolated from the wound of a patient. One month later, the patient went into septic shock and died [8]. The emergence of carbapenem-resistant bacteria is only the latest event in a process which has been observed for almost all the antimicrobials developed over the years [9]. This is particularly worrying because these antibiotics are considered one of the last resources in the treatment of multi-resistant Gram-negative infections [10,11]. Millions of lives have been saved around the world since the introduction of antibiotics in the 1940s [11,12]. Antibiotics are widely used to treat and prevent infections, including those that can occur after solid organ transplants, chemotherapy, or heart surgery [13,14,15]. In addition, in just over 100 years, the introduction of antibiotics is estimated to have contributed significantly to the increase in average human life expectancy of more than two decades [14]. Antibiotics have also played an important role in developing countries, helping to reduce morbidity and mortality from food-borne and other poverty-related infections [16,17]. However, as antibiotics have become more widely used, the problem of antibiotic resistance has begun to grow, putting humans at increasing risk [18]. In the past, resistance to penicillin and other antibiotics was successfully overcome through the discovery and development of new antibiotics. However, over time, this strategy has proved ineffective against resistant bacteria [19]. There has been a steady decline in the number of new antibiotics developed and approved over the past 30 years, likely due to economic and regulatory barriers, leaving fewer options to treat resistant bacteria [14,20]. The last original class of antibiotics was discovered in 1987. In recent years, only a few new antibiotics with limited clinical benefit have been approved [21]. The lack of antibiotics against Gram-negative bacteria is particularly worrying. Several factors contributed to the failure of the pharmaceutical industry to develop new antibiotics [22]. These include the following factors:(1)The development of new antibiotics is an extremely expensive endeavor, with a lengthy regulatory process and minimal revenues. This is because antibiotics are used for relatively short periods of time and are often curative, unlike drugs used to treat chronic diseases such as diabetes, asthma, or gastrointestinal disorders.(2)The relatively low cost of antibiotics compared to drugs used to treat neuromuscular diseases or cancer chemotherapy.(3)Lack of know-how: The research on antibiotics carried out in academia has been scaled back as a result of a lack of financial incentives due to the economic crisis.(4)Resistance can develop quickly, making it difficult to use the drug and resulting in a low return on investment for the company developing the drug [18,22].

A possible solution to the lack of antibiotic development may be to reduce the cost of the preclinical phase of the drug development process. With no guarantee that the molecule will have the desired efficacy and safety, this is the most expensive and risky phase for pharmaceutical companies [23].

Alternative methods of treating bacterial infections have also been investigated, including the use of bacteriophages and antibacterial peptides [24]. Despite their importance, these technologies suffer from major limitations which prevent them from being used in medical products [24,25]. So far, they could be a valuable addition to the antibiotics already available [25].

The aim of this review was to evaluate the literature on how these ‘titans’ contribute to the global burden of AMR. The review also highlights the need for the implementation of antimicrobial stewardship principles to prevent the transmission of drug-resistant organisms in healthcare settings. Finally, it discusses the benefits and limitations of alternative treatments for infections caused by these titans of AMR.

### 1.1. Causes and Effects of Antimicrobial Resistance

More than 65% of all antibiotics are produced by saprophytic bacteria (about 50% actinomycetes) and 20% by filamentous fungi [26]. The others are semi-synthetic, such as clindamycin (a semi-synthetic derivative of lincomycin), and synthetic, such as sulfonamides. Naturally, in line with Darwin’s theory of selection, microbes have evolved defense mechanisms against these antimicrobial substances [27]. The increase in antimicrobial resistance is due to many factors, including (1) overuse of antimicrobials, (2) inappropriate prescription of antibiotics, (3) use of antibiotics as feed additives for faster growth in livestock and poultry, (4) release of antibiotics into the environment, and (5) limited development of new antibiotics [28,29,30] (Figure 1). The direct link between overusing antibiotics and the emergence and spread of antimicrobial resistance in pathogenic bacteria has been demonstrated in several studies [31,32]. While some bacteria are naturally resistant to some antibiotics (intrinsic resistance), inappropriate use of antibiotics (e.g., taking an antibiotic for a viral infection) can promote antibiotic-resistant properties in non-pathogenic bacteria or it can allow resistant pathogens to multiply and replace non-pathogenic ones [33,34]. Unlike most animals, which inherit genes from their parents, bacteria can acquire genes from their neighbors through a process known as horizontal gene transfer (HGT) [35]. By enabling antimicrobial resistance between different species of bacteria, HGT is the main mechanism of bacterial resistance [36,37]. Despite international guidelines strongly discouraging the overuse of antibiotics, overprescribing continues worldwide [38,39]. In addition, several studies have shown that a significant percentage of prescribed therapies are inappropriate, mainly because the agent chosen or the dose/duration of treatment is not according to guidelines [40,41]. There is also growing evidence that subinhibitory concentrations of antibiotics can enhance gene transfer, biofilm formation, and quorum sensing [42,43]. There is also evidence that subinhibitory concentrations of antibiotics select for fast-growing mutants [44]. Prolonged exposure of bacterial cells to low concentrations of antibiotics can also accelerate HGT between phylogenetically distant bacteria, as well as between non-pathogenic bacteria and pathogens [45]. A direct link between subinhibitory concentrations of antibiotics in the environment and bacterial resistance acquired through selective pressure has been demonstrated in a number of studies [18,46,47]. Numerous environmental studies have shown that fertilization and irrigation with sewage sludge introduce significant amounts of antibiotics and/or their degradation products and bioactive metabolites into water and agro-ecosystems [48,49,50]. Highly water-soluble, these compounds spread rapidly in aquatic and terrestrial ecosystems [48]. They are also a source of nutrients for some microorganisms. For example, *P. aeruginosa*, which is known to be common in domestic and hospital wastewater, produces a stronger biofilm after exposure to sub-inhibitory doses of erythromycin or sulfamethoxazole [47,50]. In livestock and poultry, antibiotics are widely used as feed additives to promote faster growth. This makes the food industry a major consumer of antibiotics and a major contributor to antibiotic resistance [51,52]. It is estimated that about 70 per cent of the medically important antibiotics on the US market are intended for use in animals [53]. For many years, antibiotics have been given to animals on intensive livestock and poultry farms to help them grow faster. Although it is no longer legal to use antibiotics for this purpose, therapeutic use of antibiotics is still used to prevent the spread of disease to animals in close contact after the diagnosis of clinical disease (metaphylaxis) [54]. Antibiotics used on farms can be ingested by humans through meat products. The transfer of resistant bacteria from farm animals to humans is the result of a series of events: (1) the use of antibiotics in food-producing animals kills susceptible bacteria, leading to the development of resistant bacteria; (2) bacteria resistant to antibiotics can spread to humans both directly through contact with infected animals and indirectly through the food chain; (3) multidrug-resistant (MDR) bacteria can cause severe infections with poor outcomes in humans [2,51]. Antibiotic resistance is an old problem, dating back to the beginning of antibiotic use [12]. However, this phenomenon is now reaching crisis proportions, as the emergence of antibiotic-resistant human pathogens is far outpacing the discovery of new drugs that can provide alternative treatments [21].

### 1.2. Carbapenem Resistance in Gram-Negative Bacteria

Carbapenems, together with penicillins and cephalosporins, belong to the beta-lactam antibiotics. However, they differ from these two classes of beta-lactams in that they have an unsaturated, sulphur-free beta-lactam ring [9]. The carbapenems (imipenem, ertapenem, meropenem, and doripenem) are considered to be the treatment of last resort for infections caused by MDR organisms, defined as those not susceptible to at least one agent from three or more classes of antimicrobial agents [9,55]. Carbapenems have a unique structure that confers protection against most beta-lactamases and have concentration-independent bactericidal activity [34]. Carbapenem resistance to Gram-negative bacteria is the main contributor to multidrug resistance and is usually the last step before pan-drug resistance [56,57]. Over the past two decades, there has been an overuse of carbapenems in clinical practice on all continents to combat infections caused by an increasing number of bacterial species producing extended-spectrum beta-lactamase (ESBL), which are able to hydrolyze almost all beta-lactam antibiotics except carbapenems [57]. Carbapenems enter Gram-negative bacteria via porins [9,56]. Binding to various penicillin-binding proteins (PBPs) inhibits peptide crosslinking during cell wall synthesis, causing cell death [9]. Carbapenem resistance in Gram-negative bacteria can be attributed to several main mechanisms [58]. These include carbapenemase production, expression of efflux pumps, loss of porins, and alteration of PBPs [9]. Gram-negative bacteria like *Serratia* spp, *Pseudomonas* spp, or *Acinetobacter* spp have been found to carry carbapenemase genes on their chromosome. These bacteria would have started to produce carbapenemases under the selective pressure of antibiotics [47]. The overexpression of efflux pumps allows the bacteria to pump carbapenem out of the cells [56]. The transfer of genes encoding carbapenemases carried by mobile genetic elements (plasmids, transposons) allowed the horizontal spread of resistance genes even between different genera [35]. According to Ambler molecular classification, which is based on conserved and variable amino acid motifs of the proteins, carbapenemases belong to the classes A, B, and D [9,59]. Class A and D have a serine residue in the catalytic site. Class B is known as the metallo-β-lactamases (MBLs), because they have a metal ion (usually zinc) as a cofactor for the nucleophilic attack of the β-lactam ring. Class D enzymes are oxacillinases [59]. Class A includes chromosomal (SME, NmcA, SFC-1, BIC-1, PenA, FPH-1, SHV-38), plasmid (*Klebsiella pneumoniae* carbapenamase (KPC), GES, FRI-1), or both (IMI) encoded enzymes [56]. In general, class A carbapenemases can degrade beta-lactams (for which they have a high affinity) and carbapenems. The most important and clinically relevant class A carbapenemases are KPC and, to a lesser extent, IMI and GES [60]. Among these, the best-known KPCs have spread to all regions of the world [4]. These enzymes are generally expressed by clinically relevant organisms such as *P. aeruginosa* and *A. baumanni*. Class A types are inhibited by beta-lactamase inhibitors (clavulanate, sulbactam, tazobactam, avibactam) [61]. KPC is inhibited by boronic acid and EDTA [9]. Class B carbapenemases are metallo-β-lactamases (MBLs) with the highest carbapenemase activity and, unlike class A carbapenemases, members of this group are not inhibited by β-lactamase inhibitors [59]. EDTA and sodium mercaptoacetate can inhibit the carbapenemases in this class but cannot be used as a treatment due to their toxicity [62]. The most clinically relevant MBLs are the Verona integron-encoded MBL (VIM), imipenemase (IMP), and the New Delhi MBL [60]. Because these MBLs are usually encoded on the class 1 integron-containing gene cassettes, they spread easily among bacteria [63]. In this way, they can also integrate resistance genes that code for other classes of antimicrobial agents [56,63]. So far, 60 IMP-type carbapenemases have been described in *Enterobacteriaceae*, *Acinetobacter* spp., and *Pseudomonas* spp. VIM enzymes are among the most widespread MBLs, with >50 VIM variants reported [64]. Amongst them, VIM-2 is the most commonly reported MBL globally [65]. The *blaNDM-1* gene, encoding New Delhi metallo-beta-lactamase 1 (NDM-1), is commonly found on plasmids carrying multiple resistance genes to many antibiotics, including fluoroquinolones, aminoglycosides, macrolides, and sulfamethoxazole, resulting in extensive drug resistance [1,66]. Class D enzymes are called oxacillinases and include all OXA-type carbapenemases (e.g., OXA-48, OXA-72, and OXA-244) [9]. OXA-48 and its variants are the most important class D carbapenemases in clinical practice. Some of these enzymes can hydrolyze carbapenems (e.g., OXA-23 from *A. baumanii*) and third-generation cephalosporins (e.g., OXA-11 from *P. aeruginosa*) [67]. These enzymes are not inhibited by the classical inhibitors and play an important role in the acquired resistance of *A. baumannii* to the carbapenems [68]. OXA-type enzymes are notoriously difficult to detect because they often induce only low levels of resistance to carbapenems in vitro. However, they are among the most common carbapenemases in Gram-negative bacteria because they are associated with carbapenem treatment failure [69].

### 1.3. The Emergence of Carbapenenase-Producing Enterobacteriaceae

According to reports from the Centers for Disease Control and Prevention (CDC) and the World Health Organization (WHO), carbapenemase-producing *Enterobacteriaceae* (CPE) are by far the most pressing antimicrobial resistance threat [70]. CPE has become a major global public health threat due to difficult-to-treat CPE bacterial infections in healthcare patients [71]. *Enterobacteriaceae* are a large family of Gram-negative bacteria that includes many bacteria commonly found as part of the normal human intestinal flora [72]. Some members of this family, including *Escherichia coli*, *Klebsiella* spp., and *Enterobacter* spp., are commonly isolated from clinical cultures because of their ability to cause serious nosocomial or community bacterial infections (including septicemia, pneumonia, meningitis, and urinary tract infections) [72]. About half of all cases of sepsis and more than 70% of urinary tract infections are caused by these microorganisms [4,73]. They are also the most common cause of opportunistic infections and are also known to cause surgical site infections, abscesses, pneumonia, and meningitis [4]. A number of studies have shown that the main reservoirs of carbapenem-resistant *Enterobacteriaceae* (CRE) in healthcare facilities are colonized or infected patients, biofilms on medical devices, sink taps, and wastewater [74,75]. Prolonged intensive care unit (ICU) stay, open wound, indwelling catheter, solid organ or stem cell transplant, severe and prolonged granulocytopenia after cancer chemotherapy in critically ill patients, and prior antimicrobial therapy are also significant risk factors for acquisition of MDR *Enterobacteriaceae* [74,76]. Previous reports have shown an increasing frequency of AMR in *E. coli* and *K. pneumoniae* strains isolated from a variety of sources, including healthcare facilities, the community, and the environment [74]. There are several mechanisms by which these *Enterobacteriaceae* species can develop resistance to antibiotics. The most important are the production of ESBLs and AmpCs, the synthesis of carbapenemase enzymes, and the loss of porins [9,39]. Three main mechanisms are responsible for *Enterobacteriaceae* resistance to carbapenems: carbapenemase production, efflux pump overexpression, and porin channel mutations [77]. Of these, the production of β-lactamases capable of hydrolyzing carbapenems is the most important mechanism of resistance [58]. Mutations in porins reduce or prevent carbapenem uptake (e.g., altered expression of *ompk35* and *ompk36* in *K. pneumoniae* and loss of OmpF and OmpC in *E. coli* confer high and reduced resistance to ertapenem, respectively) [78]. By recognizing antibiotics and reducing their concentration to sub-toxic levels, drug efflux pumps play a central role in the development of multidrug resistance in *Enterobacteriaceae* [79]. Among the various efflux systems, the resistance–nodulation–division (RND) group is an important mechanism of multidrug resistance in *Enterobacteriaceae* [80]. Furthermore, the AcrAB-TolC RND, a member of the resistance–nodulation–division (RND) group, is one of the main mechanisms of multidrug resistance of *E. coli* and *K. pneumoniae* [81]. It is worth noting that inhibitors targeting this efflux pump have been shown to reverse antibiotic resistance in *Enterobacteriaceae* by restoring the efficacy of several drugs [81]. However, the main mechanism of carbapenem resistance in CRE worldwide is the production of carbapenemases such as KPC, NDM, and OXA-48-type [82]. Because these enzymes are encoded by genes carried on plasmids or other mobile genetic elements, they can be horizontally transferred to other bacterial species, making this resistance mechanism the greatest threat [35]. NDM-1 was first identified in a strain of *K. pneumoniae* from a Swedish patient in New Delhi in 2008 [83]. Since then, NDM carbapenemases have been found in *Enterobacteriaceae* isolates all over the world. Epidemiological studies indicate that intercontinental travel to endemic areas, such as India, Pakistan, and Sri Lanka, promotes the worldwide spread of clinical strains, especially *K. pneumoniae* and *E. coli*, harboring the *blaNDM-1* gene [60,84]. The presence of NDM-producing *Enterobacteriaceae* has already been reported in several European countries and worldwide [84]. KPC-producing *Enterobacteriaceae* have also been reported in many regions of the world. Epidemiological studies have shown that the United States and Europe, particularly Italy and Greece, are endemic areas for KPC-producing *Enterobacteriaceae* [85]. A total of 12 *bla_KPC_* gene variants exist globally [86]. These variants have been implicated in the outbreaks in China and the Middle East [87]. According to numerous reports, OXA-48-like carbapenemases produced by *Enterobacteriaceae* are currently spreading very rapidly worldwide [88]. However, the incidence of OXA-48-producing CPE is probably underestimated because most clinical microbiology laboratories do not test for these oxacillinases, which weakly hydrolyze carbapenems and lack cephalosporin resistance [89]. Detection of OXA-48-producing CPE must be optimized to reduce their spread for at least two important reasons: the lack of inhibition by metal ion chelators or clavulanate, and the high level of carbapenem resistance observed (in the absence of class A and B carbapenemases) when OXA-like enzymes combine with other resistance mechanisms such as ESBL and AmpC production [56,60]. *E. coli* and *K. pneumoniae* are two of the most common causes of CRE infections [11]. The mechanism of resistance to carbapenemases is often linked to the *NDM* gene, either alone or in combination with OXA-48 [82]. Class A carbapenemases have historically been susceptible to polymyxins, tigecycline, or aminoglycosides (especially gentamicin) [90]. However, resistance rates to all these drugs are steadily increasing [4]. Fortunately, the combination of ceftazidime–avibactam is effective against OXA-48 strains. This combination has better activity against the KPC and OXA-48 enzymes, but it lacks activity against the MBL [91]. Therefore, aztreonam–avibactam combination is required in the presence of the NDM resistance mechanism. A key strategy for overcoming beta-lactam resistance conferred by metallo-beta-lactamases in *Enterobacteriaceae* responsible for nosocomial infections is the combination of ceftazidime–avibactam and aztreonam [1,92].

### 1.4. The Emergence of Carbapenem-Resistant Acinetobacter baumannii (CRAB) Infections

*A. baumannii* is a Gram-negative coccobacillus that can be found throughout the environment, especially in soil and water [93]. It can also be found on the skin, in the respiratory tract, and in the gastro-intestinal tract of healthy people [93]. Carbapenem-resistant *A. baumannii* (CRAB) is an opportunistic pathogen that causes serious infections in healthcare settings [94]. The ability of *A. baumanni* to survive for long periods of time on living and non-living surfaces and its resistance to several antibiotics have made it a major public health problem worldwide, with the World Health Organization listing CRAB as a priority 1 pathogen for which new therapies are urgently needed [9,95]. CRAB rarely causes community-acquired infections, but it is emerging as a leading cause of healthcare-associated infections worldwide, including bloodstream, lung, wound, and urinary tract infections [94]. However, as humans can be colonized with this microorganism, distinguishing between colonized and infected is challenging [96]. For the same reasons, it is difficult to determine whether poor clinical outcomes are due to suboptimal antibiotic therapy or to underlying host factors (e.g., patients with acute kidney injury) [96]. In addition, CRAB infections are difficult to treat because resistance to carbapenem antibiotics is usually associated with resistance to most other antibiotics expected to be effective against the wild-type strain [97]. The main mechanisms of drug resistance in *A. baumannii* are inhibition of membrane permeability (reduction in porin permeability or increased efflux), modification of drug targets, and enzymatic inactivation of the drug by hydrolysis or formation of inactive derivatives [98]. Regardless of the mechanisms described above, *A. baumanni*, like other MDR pathogens, exhibits remarkable genetic plasticity that allows it to rapidly mutate and re-adapt [99]. The ability of *A. baumannii* to form mature biofilms on medical devices contributes significantly to both its survival under adverse environmental conditions and its exceptional antibiotic resistance [95]. Recent studies have shown that available therapies are only partially effective in reducing mortality in patients with invasive CRAB infection, which is the fourth leading cause of death attributable to antimicrobial resistance worldwide [100]. The susceptibility rates of *A. baumanni* to the carbapenemes vary according to the geographical region and are highest in Asia, Eastern Europe, and Latin America [101]. As mentioned above, although antibiotic resistance in CRAB is mediated by complex mechanisms, carbapenem resistance is commonly associated with horizontal transfer of genes encoding oxacillinase (*OXA-24/40, OXA-23*) and sometimes also metallo-β-lactamases and serine carbapenemases [102]. Previous studies have shown that the rate of resistance to key antibiotics, such as ampicillin–sulbactam and colistin, is increasing worldwide as a result of the spread of predominant CRAB clonal types [100]. In addition, the majority of CRAB infections are pneumonia, which is unresponsive to antimicrobial agents that are active against CRAB in vitro but inactive in vivo due to their poor lung penetration and dose-dependent toxicities [100]. Although the guidelines for the treatment of invasive CRAB infections differ from one organization to another (the European (ESCMID and ESICM) and the American IDSA guidelines), they all agree on the following: (a) Combination therapy with at least two agents (e.g., high-dose ampicillin–sulbactam in combination with another agent such as tigecycline, polymyxins) is recommended for the treatment of CRAB infections; (b) combination therapy with polymyxins and meropenem is not recommended; (c) the use of cefiderocol, a new FDA-approved beta-lactam with in vitro activity against CRAB isolates, should be limited to the treatment of CRAB infections refractory to other antibiotics and should be used as part of a combination regimen; (d) meropenem or high-dose imipenem–cilastatin and rifamycins or nebulized antibiotics are not recommended for the treatment of CRAB infections; (e) since polymyxin resistance develops rapidly when used as monotherapy, polymyxin B must be used in combination with at least one other agent for the treatment of CRAB infections [100,103]. Several adjunctive therapies have been proposed for the treatment of CRAB infections, but there is currently limited data to determine whether these therapies provide clinical benefit in patients with CRAB infections [96,104].

### 1.5. Emergence of Pseudomonas aeruginosa with Difficult-to-Treat Resistance

*P. aeruginosa* is an aerobic, non-fermenting, Gram-negative bacillus that is commonly associated with nosocomial infections [105]. It can cause infections in many anatomical sites, including the urinary tract, respiratory tract, soft tissues, gastrointestinal tract, and blood, especially in patients with weakened immune systems, such as those with cancer, cystic fibrosis, burns, tuberculosis, cancer, and AIDS [106]. In recent years, the presence of MDR *P. aeruginosa* isolates with limited treatment options has increased worldwide [107]. With regard to antimicrobial therapy, the definitions of MDR *P. aeruginosa* have changed in recent years, as follows. In 2008, MDR *P. aeruginosa* was defined as non-susceptibility to at least one antibiotic in at least three classes, and carbapenems were still the treatment of choice; in 2012, resistance to carbapenems increased, making the pathogen extensively drug-resistant (XDR) and pan-drug-resistant (PDR), when *P. aeruginosa* isolates also showed resistance to polymyxins, especially colistin, and tigecycline [108]. Finally, in 2018, the definition of *P. aeruginosa* with “difficult-to-treat” resistance (DTR) was adopted and defined as *P. aeruginosa* showing resistance to all of the following antibiotics: piperacillin–tazobactam, ceftazidime, cefepime, aztreonam, meropenem, imipenem–cilastatin, ciprofloxacin, and levofloxacin [109]. *P. aeruginosa* infections are often a major therapeutic challenge due to the presence of innate and acquired resistance to many antibiotics. The former involves the presence of overexpressed efflux pumps and low outer membrane permeability, while the latter is due to the acquisition or mutation of genes that contribute to resistance to several classes of antibiotics, including beta-lactams, aminoglycosides, and fluoroquinolones [110]. In recent years, there has been a gradual increase in *P. aeruginosa* isolates from healthcare-associated infections showing resistance to carbapenems, which have been one of the main treatment options for serious *P. aeruginosa* infections for about a decade [111]. Carbapenemase resistance in *P aeruginosa* can also occur in the absence of carbapenemases, i.e., through the activation of different mechanisms, such as the loss of the outer membrane porin OprD associated with the overexpression of efflux pumps or ampC [112]. While in the USA the resistance of *Pseudomonas* to carbapenems is mainly due to chromosomal genes encoding the porin OprD and the presence of specific efflux pumps, outside the USA the resistance of this pathogen to carbapenems is due to the acquisition of carbapenemases. [60,113]. In addition to carbapenemase resistance, a growing concern is the emergence of extensively drug-resistant (XDR) *P. aeruginosa* infections associated with “high-risk epidemic clones” circulating in hospitals around the world. These clones carry transmissible genetic elements that contain multiple resistance elements, including those encoding the production of selected carbapenemases and ESBLs that confer resistance to ceftolozane/tazobactam, considered the treatment of last resort for infections caused by XDR *P. aeruginosa* [112].

### 1.6. New Weapons in the War against “Titans”

In recent years, only one new antibiotic, cefiderocol, an injectable siderophore cephalosporin, has been approved to treat complicated urinary tract infections and pneumonia caused by the WHO’s most critical superbugs, including *A. baumannii, P. aeruginosa,* and *Enterobacteriaceae* [114]. Therefore, the rapid spread of resistance to new antibiotics and the slow rate of discovery of new classes of antibiotics highlight the need for innovative therapeutic antibiotic options. To reduce the risk of inducing bacterial resistance, several additions to antimicrobials are being evaluated, including nanoparticles (NPs), antimicrobial peptides (AMPs), bacteriophages, the CRISPR/Cas system, and probiotics (Table 1 and Figure 2).

#### 1.6.1. Nanoparticles

Nanoparticles are small particles between 1 and 100 nm in size. Nanoparticles are being investigated for a wide range of medical applications, from drug delivery systems and imaging agents to therapeutics [115]. Based on their composition, NPs are generally classified into three classes: organic, carbon-based, and inorganic. Of these, metallic NPs appear to be the most promising [115]. They can act directly as antibacterial agents (e.g., titanium dioxide (TiO_2_), zinc oxide (ZnO)) or as drug delivery systems (e.g., liposomes) [116]. The use of nanoparticles as a drug delivery system to target drug-resistant bacteria makes it possible to address MDR by exploiting the antibacterial activity of both Abs and NPs [116,117]. Induction of oxidative stress, release of metal ions, and non-oxidative mechanisms are the main antibacterial mechanisms of NPs. The activation of multiple mechanisms of action by NPs broadens their spectrum of antimicrobial activity and prevents the development of bacterial resistance. Previous studies have shown that NPs are effective against WHO critical priority pathogens [117].

#### 1.6.2. AMPs

AMPs are small, positively charged, amphipathic molecules, typically consisting of 12–50 amino acids. Their rapid bactericidal action, low resistance, and multifunctional mechanism of action make them one of the most promising alternatives to antibiotics [118]. The bactericidal action of AMPs involves the activation of two main different mechanisms: the depolarization and permeabilization of the bacterial membrane or the inhibition of essential intracellular functions without membrane rupture (e.g., by nucleic acid binding) [119]. A large number of antimicrobial peptides have been identified, each with a unique spectrum of activity and a different mechanism of action [119]. Although many antimicrobial peptides have been identified, clinical use of AMPs remains limited due to their limited stability and high susceptibility to protease degradation. Other obstacles include the high cost of their extraction, their low bioavailability, and their cytotoxicity [120].

#### 1.6.3. Phage Therapy

As evidenced by the many previous publications dealing with the subject, phagotherapy, i.e., the use of bacteriophages as a precision therapy for the treatment of bacterial infections, has received increasing attention over the last two decades [121]. The use of bacteriophages as antimicrobials dates back more than 100 years, and phage therapy was used worldwide until the Second World War, when the use of antibiotics gradually restricted the use of phages [122]. Increasing antibiotic resistance has led to renewed interest in bacteriophage therapy. In particular, phage therapy has been approved by the US Food and Drug Administration for the treatment of infections caused by multidrug-resistant bacteria that do not respond to available antibiotics [122]. Because bacteria can rapidly develop phage resistance, phage cocktails are highly preferred in phage therapy and have been successfully used in the treatment of life-threatening infections in humans [122]. However, further clinical research is needed to support the use of phage therapy in routine clinical practice.

#### 1.6.4. CRISPR/Cas System

The CRISPR-Cas gene editing system is used in research labs to target and eliminate plasmids that carry antibiotic resistance genes, thus preventing the spread of antibiotic resistance [123]. This technology showed immediate promise in eliminating resistance in a wide range of bacteria and has the potential to be a revolution for the future [123]. Using phage- or plasmid-based delivery vehicles, this technology has been successfully used to remove plasmids encoding gentamicin resistance genes from target bacteria [124].

#### 1.6.5. Probiotics

Several studies have shown promising results for a range of probiotics used to reduce the risk of infection and the use of antibiotics [125]. However, despite a large body of evidence showing the promising antimicrobial activity of probiotics, further studies are needed to define the doses, clinical efficacy, safety, and mechanisms of action of probiotics in humans [125,126]. This may help to reduce the development of multi-resistant bacteria.

## 2. Conclusions

Ranked by WHO as one of the top 10 global public health threats, a large and comprehensive study using data from 204 countries estimated that about 1.75 million people died from drug-resistant infections in 2019, out of 4.95 million deaths related to antimicrobial resistance, making drug-resistant infections more deadly than HIV/AIDS or malaria. This number could increase dramatically in the coming years, with AMR killing up to 10 million people a year by 2050. AMR also poses economic challenges, as it could reduce GDP by at least USD 3.4 trillion annually and push 24 million more people into extreme poverty over the next decade. Carbapenem-resistant *Enterobacteriaceae* (CRE), *A. baumannii* (CRAB), and *P. aeruginosa* (CRPA) have been identified by the WHO as critical priority bacteria for which novel therapeutics are urgently needed. These bacteria, along with those on the WHO’s priority pathogen list (vancomycin-resistant *Enterococcus faecium*, third-generation cephalosporin-resistant *Enterobacter* spp, and methicillin-resistant *Staphylococcus aureus*), belong to the group of pathogens known as ESKAPE, which are responsible for the majority of life-threatening hospital-acquired infections. CRE, CRAB, and CRPA are emerging as a major public health threat, causing healthcare-associated infections and high mortality rates due to their resistance to a wide range of antibiotics. Widespread genome sequencing has revealed that the transmission of resistance is often acquired through mobile genetic elements, including insertion sequences (IS), transposons, and conjugative plasmids. Some strains have innate resistance to carbapenems. Others contain mobile genetic elements that lead to the production of carbapenemase, which hydrolyze a broad variety of β-lactams, including carbapenems, cephalosporins, penicillin, and aztreonam. In addition, the co-localization of carbapenemase production genes with other resistance genes in these pathogens further limits the treatment options for patients. Although the development of resistance in the microorganism occurs naturally, the misuse and overuse of antimicrobials in human health, food animal production, and agriculture are the main cause of antimicrobial resistance. In addition, several studies have shown that consumption of contaminated food and improper food handling can lead to human exposure to antimicrobial resistance. There have been numerous calls for strategies designed to prevent the further development and spread of AMR, including the World Health Organization’s (WHO) approval of the Global Action Plan on Antimicrobial Resistance in 2015. In 2019, AMR has been identified as one of the top 10 public health threats facing humanity. Given the high prevalence, associated morbidity and mortality, and limited treatment options, we have focused on CRE-CRAB-CRPA infections, which also have the potential to cause hospital outbreaks worldwide and contribute to the spread of resistance. In addition, CRE-CRAB-CRPA infections have been shown to be preceded by colonization in almost all cases. Therefore, early diagnosis of colonization with CRE-CRAB-CRPA could most likely help to identify patients most at risk of developing infection. Surveillance of CRE-CRAB-CRPsA infection is essential and consists of identifying carbapenemase resistance in CRE-CRAB-CRPsA isolates to prevent transmission of these pathogens to other patients. There is a need for alternative treatment options for bacterial infections. We cannot rely on antibiotics alone. It is important to understand that the discovery of one or a few new antibiotics will not be the ‘solution’ to antibiotic resistance. However, since we will still need antibiotics in the short and medium term, it is important to learn from past mistakes in order to preserve every new antibiotic that comes onto the market. While it is true that there is potentially no antibiotic to which bacteria cannot become resistant, it is also true that using them carefully will slow down the process. This means that as new antibiotics come onto the market, they must be used wisely, or they will quickly lose their effectiveness as bacteria develop resistance.

## Figures and Tables

**Figure 1 microorganisms-11-01912-f001:**
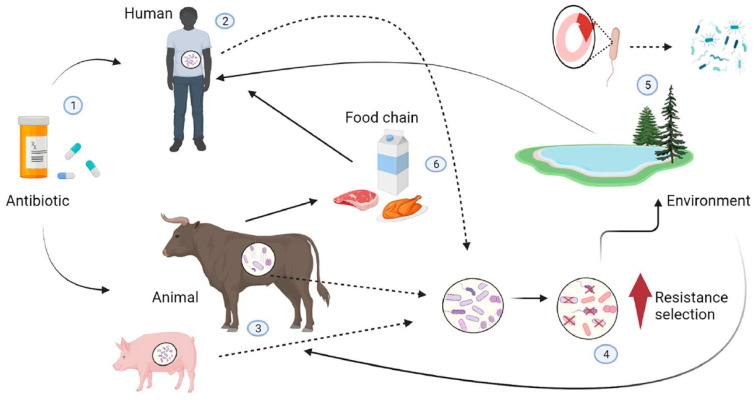
Transmission pathways for resistant bacteria between food animals, humans, and the environment. The inappropriate use of antibiotics (1) in humans (2) (due to inappropriate prescribed therapy) and animals (3) (used as feed additives) can create a selective pressure favoring antibiotic resistance properties in non-pathogenic bacteria or allow resistant pathogens to proliferate and replace nonpathogenic ones in animals and humans (4). Resistant bacteria and their genes can reach the environment (5), which acts as a reservoir where mobile genetic elements carrying resistance genes are exchanged with the bacterial flora of the environment by horizontal gene transfer and spread to other human and animal hosts by various routes (soil, water, air). Resistant bacteria can also spread from animals to humans through the food chain (6).

**Figure 2 microorganisms-11-01912-f002:**
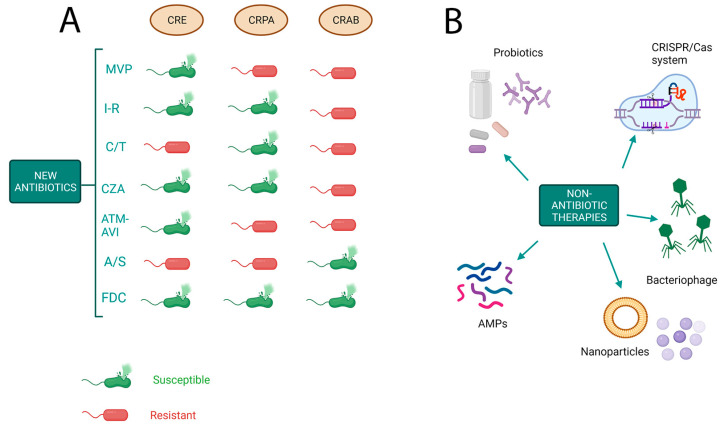
Novel strategies to tackle antimicrobial resistance. (**A**) Activity of new antibiotics against Gram-negative bacteria on the basis of resistance mechanisms. CRE: carbapenem-resistant *Enterobacteriaceae*; CRPA: carbapenem-resistant *Pseudomonas aeruginosa*; CRAB: cabapenem-resistant *Acinetobacter baumannii*; MVP: meropenem–vaborbactam; I-R: imipenem–relebactam; C/T: ceftalozane–tazobactam; CZA: ceftazidime–avibactam; ATM-AVI: aztreonam–avibactam; A/S, ampicillina–sulbactam; FDC, cefiderocol. (**B**) Novel approaches to treat multidrug-resistant bacteria. AMPs: antimicrobial peptides.

**Table 1 microorganisms-11-01912-t001:** Novel control strategies to tackle AMR Pathogens.

Newer Approaches	Reference
nanoparticles	[115,116,117]
antimicrobial peptides	[118,119,120]
Bacteriophages	[121,122]
CRISPR/Cas system	[123,124]
probiotics	[125,126]

## Data Availability

Not applicable.

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
