# Peer review of "The Challenge of Overcoming Antibiotic Resistance in Carbapenem-Resistant Gram-Negative Bacteria: “Attack on Titan”"

_microorganisms, 2023, doi:10.3390/microorganisms11081912_

Round 1
Reviewer 1 Report
The paper deals with the antimicrobial resistance (AMR) of bacteria, focusing on carbapenem-resistant gram-negative bacteria. In the Introduction section, the authors gave information on strategies and regulatory policies to control the spread of AMR. Further, they discussed carbapenem-resistant gram-negative bacteria as one of the major urgent threats to public health. Finally, they reviewed the advantages and disadvantages of alternative therapies for controlling bacterial-resistant strains.
The paper is well-written and organized, reviewing the most recent literature sources. After minor revisions, it should be considered for publication.
List of drawbacks:
- The title should be changed. for exp. Carbapenem-resistant gram-negative bacteria. The title in the present form is not in accordance with the manuscript.
- Please consider that lines 119-153 could be a part of the Introduction section (at the end of the paragraph) better than in 1.1: Causes and Effects of Antibiotic Resistance
- At the end of the Introduction section, add the aim/aims.
- Lines 29, 31,34,36, 68,74,75 Decide on antibiotic resistance or antimicrobial resistance. Antimicrobials include antibiotics and related semi-synthetic or synthetic antimicrobial agents.
- Line 34: Instead of ref. 3, it should be ref. O'Neill J. Review on Antimicrobial Resistance Antimicrobial Resistance: Tackling a crisis for the health and wealth of nations. London: 2014.
- Line 40 Instead of ref. 5, it should be Global Action Plan on Antimicrobial Resistance. Geneva: World Health Organization; 2015.
- Lines 41, 43: Reference 7 is not necessary.
- Line 91: Please explain the statement that subinhibitory concentrations of antibiotics can enhance antibiotic production.
- Lines 32, 36, 115, 159, 166, 181, 191, 230, 231, 394: The abbreviations should be defined at their first appearance in the text.
- The section Conclusion should not contain references.
- The Latin names of bacterial species, in vitro, in vivo, genes should be in italics.
Author Response
We would like to thank the reviewers since their comments have greatly helped us to improve the manuscript which was carefully revised according to their suggestions.
Reviewer n.1
The title should be changed. for exp. Carbapenem-resistant gram-negative bacteria. The title in the present form is not in accordance with the manuscript
Thanks for this helpful suggestion. Title changed as suggested by referee
Please consider that lines 119-153 could be a part of the Introduction section (at the end of the paragraph) better than in 1.1: Causes and Effects of Antibiotic Resistance
We have moved the lines to the end of the Introduction, in agreement with the reviewer (lines 67-90).
At the end of the Introduction section, add the aim/aims.
Thanks for this helpful suggestion. The aims have been added at the end of the introduction (lines 91-95)
Lines 29, 31,34,36, 68,74,75 Decide on antibiotic resistance or antimicrobial resistance. Antimicrobials include antibiotics and related semi-synthetic or synthetic antimicrobial agents.
Thanks for this helpful suggestion. We have changed antibiotic resistance to antimicrobial resistance (lines 29, 34, 98, 108).
Line 34: Instead of ref. 3, it should be ref. O'Neill J. Review on Antimicrobial Resistance Antimicrobial Resistance: Tackling a crisis for the health and wealth of nations. London: 2014.
Thank you for nothing. The correct reference has been added to the revised manuscript.
Line 40 Instead of ref. 5, it should be Global Action Plan on Antimicrobial Resistance. Geneva: World Health Organization; 2015
Reference 5 has been changed as suggested (see reference 6 of the revised manuscript).
Lines 41, 43: Reference 7 is not necessary
Reference 7 has been removed
Please explain the statement that subinhibitory concentrations of antibiotics can enhance antibiotic production.
There was a mistake in the statement. The whole sentence has been rewritten (lines 122-123). Thank you for noticing.
Lines 32, 36, 115, 159, 166, 181, 191, 230, 231, 394: The abbreviations should be defined at their first appearance in the text.
Thank you for noticing. Now, the abbreviations are defined on first use.
The section Conclusion should not contain references.
Thank you for noticing. References have been deleted
The Latin names of bacterial species, in vitro, in vivo, genes should be in italics.
Thank you for pointing this out. We have carefully checked the entire manuscript
Reviewer 2 Report
The paper is very well written. There are just two recommendations: 1) change the name Enterobacteriaceae into correct one which is Enterobacterales; 2) on the page 9. line 388 - I would recommend use term 'addition to antimcirobials' rather then 'alternatives'.
Author Response
The paper is very well written. There are just two recommendations: 1) change the name Enterobacteriaceae into correct one which is Enterobacterales; 2) on the page 9. line 388 - I would recommend use term 'addition to antimcirobials' rather then 'alternatives'.
We thank the referee for the careful and insightful review of our manuscript. The recommendations listed have been complied with
Reviewer 3 Report
The manuscript by Mancuso et al., discusses antimicrobial resistance in gram negative bacteria, with particular focus on MDR and resistance to carbapenems.
AMR is one of the priority concerns for the health organisations worldwide and a major threat to public health. In recent years dozens (if not hundreds) of reviews discussing AMR have been published. The authors of this manuscript however too a slightly different approach and combined a review of the mechanisms of AMR in some of the bacteria from the WHO priority list of pathogens to which new antimicrobials are urgently needed, and the available or novel approaches to treat these pathogens.
I have a several comments/suggestions that should be applied by the authors.
General comments:
The authors reference multiple reviews instead of the primary research articles. This means the reader of this review would need to read another review to find out the actual source of the data. E.g. ref 10, 11, 12, 13, 14 and so on, are all reviews and majority of these must be replaced by the original source.
The authors should be consistent when writing the names of genes
Please write a full species name e.g. Klebsiella pneumoniae only when used for the first time
The authors should divide large blocks of text into shorter paragraphs concerning a specific argument each
Specific comments
Title - the curious title can definitely attract attention, although I am not sure if the reference to a manga is accidental or not. However, in the manuscript I no longer see a reference to the described pathogens as 'titans' or to an 'attack on titans' in general. I would suggest renaming the section 1.6. to fir the title and maybe instead of superbugs use the term 'titans'. Otherwise the authors can remove 'attack on titans' from the title.
Abstract: L 14 a word is missing after 'drug resistance' - genes? traits?
Introduction: L32 - L34 - the data about 10 mln people dying from infections caused by MDR bacteria has never truly been supported by the research evidence. Although it has been quoted extensively especially by the various AMR campaigns, it should not be placed in a scientific review - please see de Kraker et al., 2016 PloS Med.
L52 - authors often refer to carbapenems as last resort antimicrobials, which may be true in some cases, but not all. More often polymyxins are treated as the last resort antimicrobials for MDR gram negative infections, which exception of pathogens with intrinsic resistance.
1.1
L73-77 The factors can lead to the increased rates of AMR acquisition but are not the main 'aetiology' .
L80 - probably 'inappropriate use' rather than overuse
L81 - please avoid using word 'harmless' throghout the manuscript as it has no real significance here. Depending on the context it can be replaces by colonizing, microbiota, non pathogenic, susceptible to antimicrobials etc.
L82 'Unlike most animals, which inherit genes from their parents, bacteria can acquire genes from non-relatives through a process known as horizontal gene transfer (HGT)' - please change this sentence as it can be understood as if bacteria belong to Animal kingdom and have relatives..
L85 - antibiotic resistance 'gene transfer' ?; HGT is not a cause, it is a mechanism
L91 - what do authors mean by 'antibiotic production' ?
L94 - do authors mean sub-inhibitory concentrations? the phrase 'low levels' is measleading
L102 - what does it mean consistent biofilm? more homogeneous?
L110 - do authrs refer to metaphylaxis? - please use appropriate terms
L112-115 - this part is oversimplified. The authors should rephrase it and attempt to to give much higher impact of the One Health concept.
L152 - The last sentence is unclear
1.2
L155 - please avoid using word 'like'
L164-167 - please rephrase the sentence as it is unclear
L172 - 'have carbapenemases genes in their chromosomes' - please re-write in a scientific way
L174 - Changes .. - sentence unclear
L187 - if talking about KPC (Klebriella pneumoniae carbapenamase) the authors should probably mention K. pneumoniae at the first place
1.3
L235 - replace resistance with AMR
L237-240 - Please re-write as the sentence is not clear
L241 - are the authors sure it is 'formation' of efflux pumps?
1.4
L307 - replace word followed
L312 - why 'unique'? genome of many MDR bacteria including Enterobacteria are generally plastic.
1.5
L370 Please re-write- the sentence is misleading
L374 - Reference 111 doesn't seem correct for this data
1.6. Please make sure to divide the section into appropriate paragraphs.
Authors should prepare a table with the novel ways to treat or prevent AMR/MDR - adding appropriate references
L425 - what do authors mean by 'block the development of AMR' please provide more details.
The manuscript should be proofread for language mistakes
Author Response
We would like to thank the reviewer since his/her comments have greatly helped us to improve the manuscript which was carefully revised according to his/her suggestions.
The authors reference multiple reviews instead of the primary research articles. This means the reader of this review would need to read another review to find out the actual source of the data. E.g. ref 10, 11, 12, 13, 14 and so on, are all reviews and majority of these must be replaced by the original source.
We thank the referee for the careful and insightful review of our manuscript. We have added the original sources as requested, but we prefer, if the reviewer agrees, to keep the reviews as well, which can give the reader a broader picture of the subject.
The authors should be consistent when writing the names of genes. Please write a full species name e.g. Klebsiella pneumoniae only when used for the first time
Thank you very much for this helpful suggestion. We have carefully reviewed the entire manuscript
The authors should divide large blocks of text into shorter paragraphs concerning a specific argument each
Thanks for this helpful suggestion. In the revised manuscript, the text has been divided into shorter paragraphs. We have also moved lines 119-153 to the end of the Introduction, as requested by reviewer #1.
Title - the curious title can definitely attract attention, although I am not sure if the reference to a manga is accidental or not. However, in the manuscript I no longer see a reference to the described pathogens as 'titans' or to an 'attack on titans' in general. I would suggest renaming the section 1.6. to fir the title and maybe instead of superbugs use the term 'titans'. Otherwise the authors can remove 'attack on titans' from the title
Thank you for this helpful suggestion. In the section 1.6 the term superbugs was replaced with “titans”
Abstract: L 14 a word is missing after 'drug resistance' - genes? traits?
Thank you for noticing this. The missing word is “genes”
Introduction: L32 - L34 - the data about 10 mln people dying from infections caused by MDR bacteria has never truly been supported by the research evidence. Although it has been quoted extensively especially by the various AMR campaigns, it should not be placed in a scientific review - please see de Kraker et al., 2016 PloS Med.
We agree with the referee. We have changed the sentence accordingly (lines 32-34).
L52 - authors often refer to carbapenems as last resort antimicrobials, which may be true in some cases, but not all. More often polymyxins are treated as the last resort antimicrobials for MDR gram negative infections, which exception of pathogens with intrinsic resistance.
We agree with the referee. We have changed the sentence accordingly (line 52).
L73-77 The factors can lead to the increased rates of AMR acquisition but are not the main 'aetiology'
We agree with the referee. We have changed the sentence accordingly (lines 103-104)
L80 - probably 'inappropriate use' rather than overuse
We agree with the referee. We have changed the sentence accordingly (line 110)
L81 - please avoid using word 'harmless' throghout the manuscript as it has no real significance here. Depending on the context it can be replaces by colonizing, microbiota, non pathogenic, susceptible to antimicrobials etc.
We agree with the referee and have therefore replaced the word "harmless" throghout the manuscript.
L82 'Unlike most animals, which inherit genes from their parents, bacteria can acquire genes from non-relatives through a process known as horizontal gene transfer (HGT)' - please change this sentence as it can be understood as if bacteria belong to Animal kingdom and have relatives.
Thank you for this helpful suggestion. The indicated sentence was changed (line 113).
L85 - antibiotic resistance 'gene transfer' ?; HGT is not a cause, it is a mechanism.
We agree with reviewer and replaced the word cause with mechanism (line 116)
L91 - what do authors mean by 'antibiotic production' ?
This point was also raised by referee no. 1. There was an error in the statement. The whole sentence has been rewritten (lines 121-123)
L94 - do authors mean sub-inhibitory concentrations? the phrase 'low levels' is measleading
We agree with Reviewer and changed the phrase (lines 125-126)
L102 - what does it mean consistent biofilm? more homogeneous?
We have rephrased the sentence to make it clearer (lines 132-133)
L110 - do authrs refer to metaphylaxis? - please use appropriate terms
Thank you for this helpful suggestion. We have rephrased the sentence to make it clearer (lines 140-142)
L112-115 - this part is oversimplified. The authors should rephrase it and attempt to to give much higher impact of the One Health concept.
Thank you for this helpful suggestion. This part has been rephrased to make it clearer (lines 144-148).
L152 - The last sentence is unclear
This sentence has been changed (lines 89-90)
L155 - please avoid using word 'like'
Thank you for this helpful suggestion. The word like has been deleted (line 165).
L164-167 - please rephrase the sentence as it is unclear
Thank you for this helpful suggestion. The sentence has been rephrased (lines 174-177)
L172 - 'have carbapenemases genes in their chromosomes' - please re-write in a scientific way
Thank you for this helpful suggestion. The sentence has been rewritten (lines 183-184)
L174 - Changes .. - sentence unclear
The sentence has been changed (line 185)
L187 - if talking about KPC (Klebriella pneumoniae carbapenamase) the authors should probably mention K. pneumoniae at the first place
Thank you for this helpful suggestion (line 194)
L235 - replace resistance with AMR
Done (line 246)
L237-240 - Please re-write as the sentence is not clear
Thank you for this helpful suggestion. The sentence has been rephrased (lines 248-250)
L241 - are the authors sure it is 'formation' of efflux pumps?
The reported error has been corrected. Thank you for noticing (line 252)
L307 - replace word followed
The word followed has been changed with “associated” (line 317)
L312 - why 'unique'? genome of many MDR bacteria including Enterobacteria are generally plastic.
We agree with Reviewer. The sentence has been rephrased (line 322-323)
L370 Please re-write- the sentence is misleading
The sentence has been changed (line 381-383)
L374 - Reference 111 doesn't seem correct for this data
The indicated reference has been changed (line 383)
1.6. Please make sure to divide the section into appropriate paragraphs.
Authors should prepare a table with the novel ways to treat or prevent AMR/MDR - adding appropriate references
Thank you for this helpful suggestions. The section has been divided into appropriate paragraphs and a new table has been added to the revised manuscript
L425 - what do authors mean by 'block the development of AMR' please provide more details.
Thank you for this helpful suggestions. More details have been added in the revised manuscript (lines 442-444).
The manuscript has been thoroughly proof-read and it is our sincere hope that it will be to your approval.
Round 2
Reviewer 3 Report
I would like to thank the authors for implementing the suggested changes in the new version of the manuscript.
I would like to mention that the change of Enterobactericeae to Enterobacterales was not necessary, as the paragraph describes the members of the family rather than the order. The acronym CRE most frequently refers to Enterobactericeae as you can see e.g. n the reports of EFSA: https://www.ecdc.europa.eu/en/publications-data/directory-guidance-prevention-and-control/prevention-and-control-infections-1 and WHO https://www.who.int/publications/i/item/9789241550178
I believe using Enterobacterales is a too broad category, as the authors do not write about many members of the order such as Morganella spp, Serratia spp, Hafnia spp., which carry multiple intrinsic resistances to beta lactam antibiotics.
If the authors decide to keep the order name, they should change line 232 into 'order' and not family, and at least list some of the more important human pathogens of the order and their main molecular mechanisms of resistance.
The sentence 'Carbapenems are a group of beta-lactam antibiotics such as penicillins and cephalosporins' is still confusing. I would replace it with: 'Carbapenems, together with penicillins and cephalosporins, belong to the beta-lactam antibiotics'
Some minor mistakes are still present in the manuscript, see L 397
Some minor errors present
Author Response
We thank the reviewer for his/her careful comments and efforts to improve our manuscript.
We agree with the reviewer that Enterobactericeae should be retained as the paragraph describes the family members and not the order. We have made the appropriate changes to the manuscript.
The sentence "Carbapenems are a group of beta-lactam antibiotics such as penicillins and cephalosporins" has been amended as suggested by the Referee (lines 165-166).
The mistake has been corrected (line 397). Thanks for noticing.
The manuscript has been thoroughly proof-read and we hope that the revised version of the manuscript can now meet the reviewers’ expectations and can be accepted for publication.